

# Ideas and perspectives: Errors associated with the gross nitrification rates in forested catchments calculated from the triple oxygen isotopic composition (Δ$^{17}$O) of stream nitrate

Weitian Ding[1], Urumu Tsunogai[1], Fumiko Nakagawa[1]

[1]Graduate School of Environmental Studies, Nagoya University, Furo-cho, Chikusa-

ku, Nagoya 464-8601, Japan

Corresponding author: Weitian Ding (ding.weitian.v2@s.mail.nagoya-u.ac.jp)



**Abstract**
A novel method to quantify the gross nitrification rate (GNR) in each forested
catchment using the triple oxygen isotopic composition ($\Delta^{17}O$) of stream nitrate eluted
from the catchment has been proposed and used in recent studies. However, the
equations used in the calculations assumed homogeneous $\Delta^{17}O$ values of nitrate being
metabolized through either assimilation or denitrification within the forested soil
layers without particular discussions. The GNR estimated from the $\Delta^{17}O$ of stream
nitrate using the equations was more than six times the actual GNR in our simulated
calculation for a forested catchment where the $\Delta^{17}O$ values of nitrate being
metabolized in the soil were heterogeneous and showed a decreasing trend with
increasing depths. Therefore, we should verify that the $\Delta^{17}O$ values of nitrate being
metabolized are homogeneous in forested soils or estimate the possible range of errors
using $\Delta^{17}O$ of stream nitrate to estimate the GNR.
**1 Introduction**
Nitrate ($NO_3^-$) is a crucial nutrient in forest ecosystems that often limits primary
production. Nitrification is the microbial process that generates $NO_3^-$ from the
ammonium in a forested ecosystem; therefore, the nitrification rate is an important
parameter to be quantified when evaluating each forest ecosystem's present and future
states. The total rate of $NO_3^-$ production, gross nitrification rate (GNR), reflects
internal N cycling better than the net nitrification rate (Bengtsson et al., 2003),



especially in forest ecosystems where the GNR often exceeds the net nitrification rate
by order of magnitude (Stark and Hart, 1997; Verchot et al., 2001).
Recently, several studies successfully determined GNR in each water environment
using the $\Delta^{17}O$ values of $NO_3^-$ as a conserved tracer for the mixing ratio between the
atmospheric nitrate ($NO_3^-{}_{atm}$) deposited into each water environment and
remineralized nitrate ($NO_3^-{}_{re}$) produced through nitrification therein (Tsunogai et al.,
2011, 2018). Although $NO_3^-{}_{re}$ always has a $\Delta^{17}O$ value close to 0 ‰ because its
oxygen atoms come from either terrestrial $O_2$ or $H_2O$ through nitrification, $NO_3^-{}_{atm}$
displays an anomalous enrichment in $^{17}O$ with a $\Delta^{17}O$ value being approximately
+26 ‰ (Tsunogai et al., 2010, 2016) because of oxygen transfers from atmospheric
ozone (Michalski et al., 2003; Nelson et al., 2018). Additionally, $\Delta^{17}O$ is almost stable
during "mass-dependent" isotope fractionation processes (Michalski et al., 2004;
Tsunogai et al., 2016); therefore, regardless of partial metabolism through
denitrification or assimilation after deposition, $\Delta^{17}O$ can be used as a conserved tracer
of $NO_3^-{}_{atm}$ to calculate the mixing ratio of $NO_3^-{}_{atm}$ within total $NO_3^-$
($NO_3^-{}_{atm}/NO_3^-{}_{total}$) using the following equation:
$$[NO_3^-{}_{atm}]/[NO_3^-{}_{total}] = [NO_3^-{}_{atm}]/([NO_3^-{}_{re}] + [NO_3^-{}_{atm}]) = \Delta^{17}O_{water}/\Delta^{17}O_{atm} \qquad (1)$$
where $\Delta^{17}O_{atm}$ and $\Delta^{17}O_{water}$ denote the $\Delta^{17}O$ values of $NO_3^-{}_{atm}$ and $NO_3^-$ dissolved in
each water environment, respectively. Using both the $NO_3^-{}_{atm}/NO_3^-{}_{total}$ ratio estimated
from the $\Delta^{17}O$ value of $NO_3^-$ in a lake water column and the deposition rate of



$NO_3^-{}_{atm}$ into the lake, past studies successfully estimated GNR therein (Tsunogai et
al., 2011, 2018).
In addition to water environments, the $\Delta^{17}O$ method has been further applied to
forested catchments to determine GNR (Fang et al., 2015; Hattori et al., 2019; Huang
et al., 2020; Riha et al., 2014). By using the deposition flux of $NO_3^-{}_{atm}$ into the
catchment as well as the elution flux of both unprocessed $NO_3^-{}_{atm}$ and $NO_3^-{}_{re}$ via
stream, which can be determined from the $\Delta^{17}O$ values of $NO_3^-$ in stream water eluted
from the catchment, GNR in each forested catchment has been determined in a
manner similar to the water environments (Fang et al., 2015). Applying the $\Delta^{17}O$
method to forested soils, where the $\Delta^{17}O$ values of $NO_3^-$ are often heterogeneous
(Costa et al., 2011; Hattori et al., 2019), should be done with extreme caution, in
contrast to water environments where the $\Delta^{17}O$ values of nitrate were largely
homogeneous. We present an accurate relationship between the $\Delta^{17}O$ and GNR
starting from the basic isotope mass balance equations to explain the problem of using
the $\Delta^{17}O$ method in such heterogeneous environments.

**2 Calculation**
The total mass balance equation of $NO_3^-$ including GNR in each catchment can be
expressed as follows:
$NO_3^-{}_{deposition} + GNR = NO_3^-{}_{leaching} + NO_3^-{}_{uptake} + GDR$         (2)



where $NO_3^-{}_{deposition}$, GNR, $NO_3^-{}_{leaching}$, $NO_3^-{}_{uptake}$, and GDR denote the deposition flux
of $NO_3^-$ into each catchment, the gross nitrification rate in each catchment, the
leaching flux of $NO_3^-$ from each catchment, the uptake rate of $NO_3^-$ in each
catchment, and the gross denitrification rate in each catchment, respectively.
The isotope mass balance for each $\Delta^{17}O$ value of $NO_3^-$ in the catchment can also be
calculated using the same method:
$NO_3^-{}_{deposition} \times \Delta^{17}O(NO_3^-)_{atm} + GNR \times \Delta^{17}O(NO_3^-)_{nitrification} = NO_3^-{}_{leaching} \times \Delta^{17}O(NO$
$_3^-)_{stream} + NO_3^-{}_{uptake} \times \Delta^{17}O(NO_3^-)_{uptake} + GDR \times \Delta^{17}O(NO_3^-)_{denitrification}$      (3)
where $\Delta^{17}O(NO_3^-)_{atm}$, $\Delta^{17}O(NO_3^-)_{nitrification}$, $\Delta^{17}O(NO_3^-)_{stream}$, $\Delta^{17}O(NO_3^-)_{uptake}$, and
$\Delta^{17}O(NO_3^-)_{denitrification}$ denote the $\Delta^{17}O$ value of $NO_3^-{}_{atm}$ deposited in each catchment,
that of $NO_3^-{}_{re}$ produced through nitrification, that of $NO_3^-$ eluted from each
catchment, that of $NO_3^-$ assimilated by plants and other organisms in each catchment,
and that of $NO_3^-$ decomposed through denitrification in each catchment, respectively.
If the $\Delta^{17}O$ values of $NO_3^-$ were homogeneous in forested soils where $NO_3^-$ was
metabolized through either assimilation (by plants and other organisms) or
denitrification, Eq. 4 can be expressed as follows:
$\Delta^{17}O(NO_3^-)_{stream} = \Delta^{17}O(NO_3^-)_{uptake} = \Delta^{17}O(NO_3^-)_{denitrification}$      (4)
Consequently, by combining Eqs. 3 and 4, we could obtain the following
relationship:
$NO_3^-{}_{deposition} \times \Delta^{17}O(NO_3^-)_{atm} + GNR \times \Delta^{17}O(NO_3^-)_{nitrification} = (NO_3^-{}_{leaching} + NO_3^-{}_{uptak}$
$_e + GDR) \times \Delta^{17}O(NO_3^-)_{stream}$      (5)





We could estimate GNR using Eq. 6 obtained from Eqs. 2 and 5 because we can
approximate the $\Delta^{17}O$ values of $NO_3^-{}_{re}$ produced through nitrification
($\Delta^{17}O(NO_3^-)_{nitrification}$) to be 0 (Michalski et al., 2003; Tsunogai et al., 2010):
$GNR = NO_3^-{}_{deposition} \times (\Delta^{17}O(NO_3^-)_{atm} - \Delta^{17}O(NO_3^-)_{stream})/\Delta^{17}O(NO_3^-)_{stream}$     (6)
The Eq. 6 corresponds to that used in previous studies for quantifying GNR in each
forested catchment (Fang et al., 2015; Hattori et al., 2019; Huang et al., 2020; Riha et
al., 2014).

**91     3 Results and Discussion**

The $\Delta^{17}O$ values of $NO_3^-$ metabolized in each catchment should be homogeneous
and therefore correspond with those of $NO_3^-$ in the stream, as presented in Eq. 4 to
obtain Eq. 6. However, many of the forested catchments do not satisfy this condition
needed to obtain Eqs. 4–6. In the studied forested soils, Hattori et al. (2019) found a
decreasing trend in the $\Delta^{17}O$ values of $NO_3^-$ together with the depth, from more than
+20‰ at the surface soil to less than +3‰ at depths of 25 to 90 cm from the soil
surface. Furthermore, most of the $\Delta^{17}O$ values of soil $NO_3^-$ differed from those in the
stream eluted from the catchment (+2.2‰ on average) in Hattori et al. (2019).
To demonstrate the possible change in GNR per the variation in the $\Delta^{17}O$ values of
$NO_3^-$ in forested soils, we estimated GNR for two simulated forested soils: that with a
vertically heterogeneous $\Delta^{17}O$ of $NO_3^-$ (Fig. 1a) and that with a vertically
homogeneous $\Delta^{17}O$ of $NO_3^-$ (Fig. 2a). Because Hattori et al. (2019) reported the





$NO_3^-{}_{deposition}$ to be 7.0 kg of N ha$^{-1}$ y$^{-1}$, $NO_3^-{}_{leaching}$ to be 2.6 kg of N ha$^{-1}$ y$^{-1}$,
$\Delta^{17}O(NO_3^-)_{atm}$ to be +28.0 ‰, and $\Delta^{17}O(NO_3^-)_{stream}$ to be +2.2 ‰ in the forested
catchment, we adopted the same parameter in the simulated calculation in this study.

107        We divided the soils in the heterogeneous forest soils into 10 layers in the vertical

direction simulating the soils observed by Hattori et al. (2019), where the $\Delta^{17}O$ values
of $NO_3^-$ gradually decreased with increasing depths, showing the $\Delta^{17}O$ values from
+28.0 to +2.2 ‰ with a decrease rate of +2.58 ‰ for each step (Fig. 1b). Similarly,
we assumed gradual decrease with increasing depths in the leaching flux of $NO_3^-$, i.e.,
from 7 to 2.6 kg of N ha$^{-1}$ y$^{-1}$ with a decrease rate of 0.44 kg of N ha$^{-1}$ y$^{-1}$ for each
step (Fig. 1c). In the homogeneous forest soils, we also divided the forested soils into
10 layers in the vertical direction. The vertical changes in the leaching flux of $NO_3^-$
were the same as those in the heterogeneous soils (Fig. 2c), whereas the $\Delta^{17}O$ values
of $NO_3^-$ were constant to be +2.2 ‰ in the soil layers (Fig. 2b).

117        Applying the total mass balance and isotope mass balance of $NO_3^-$ shown in Eqs. 2

and 3 to each layer, we estimated both GNR (Figs. 1e and 2e) and total metabolic rate
of $NO_3^-$ (GDR + uptake) (Figs. 1d and 2d) in each layer assuming that: (1) the $\Delta^{17}O$
values of $NO_3^-$ were constant in each layer, (2) the vertical flow of $NO_3^-$ in the soil
layers was only downward from surface to the water layer with a uniform residence
time in each layer, and (3) the GNR and metabolic rate of $NO_3^-$ (GDR + uptake) was
zero in the water layer (the layers beyond the no. 10 soil layer). Then, by integrating
the GNR determined for each layer, we can estimate the total GNR in each forested





catchment. Although the GNR simulated for the catchment with the homogeneous
$\Delta^{17}O$ values of $NO_3^-$ in the forested soils showed a value of 83.6 kg of N ha$^{-1}$ y$^{-1}$
equal to that estimated by Hattori et al. (2019) (Fig. 2e), the total GNR became a
much smaller value of 13.0 kg of N ha$^{-1}$ y$^{-1}$ simulated for the catchment with the
heterogeneous $\Delta^{17}O$ values of $NO_3^-$ in the forested soils (Fig. 1e). As a result, we
conclude that the distribution of the $\Delta^{17}O$ values of $NO_3^-$ in the forested soils can
significantly affect the overall GNR in forested catchments as calculated from the
$\Delta^{17}O$ of stream $NO_3^-$.

133       If we estimated the downward water flux at each soil layer, together with the

concentration and $\Delta^{17}O$ value of $NO_3^-$ in each soil layer using a tension-free lysimeter
(Inoue et al., 2021), we could estimate the vertical changes in the leaching flux of
$NO_3^-$ for each soil layer together with the $\Delta^{17}O$ value of each $NO_3^-$. Then, applying
the mass balance and isotope mass balance shown in Eqs. 2 and 3 in each layer, we
can estimate a more accurate GNR of the forested catchment by integrating the GNR
estimated for each soil layer together with the more accurate metabolic rate of $NO_3^-$
(GDR + uptake) of the forested catchment. However, without such observation on the
distribution of the $\Delta^{17}O$ value of $NO_3^-$, it is difficult to assume that the $\Delta^{17}O$ values of
$NO_3^-$ were homogeneous in forested soils where $NO_3^-$ was metabolized, so that the
GNR should be reported with errors in which possible variations in the $\Delta^{17}O$ values of
soil $NO_3^-$ have been considered.



**4 Conclusion**


Past studies proposed the $\Delta^{17}O$ method to determine GNR in each forested
catchment. The equations used in the calculation presupposed that the $\Delta^{17}O$ values of
$NO_3^-$ in forested soils were homogeneous, however, they are often heterogeneous and
showing a decreasing trend with increasing depths. It must be essential to
clarify/verify the distribution of the $\Delta^{17}O$ values of $NO_3^-$ in forested soils before
applying the $\Delta^{17}O$ values of stream $NO_3^-$ to estimate GNR.

*Data availability.* All data are presented in the Supplement.

*Author contributions.* WD, UT, and FN designed the study. WD and UT performed
data analysis and wrote the paper.

*Competing interests.* The authors declare that they have no conflict of interest.

*Acknowledgments*
The authors are grateful to the members of the Biogeochemistry Group, Nagoya
University, for their valuable support throughout this study. This work was supported
by a Grant-in-Aid for Scientific Research from the Ministry of Education, Culture,
Sports, Science, and Technology of Japan under grant numbers 22H00561,
17H00780, 22K19846, the Yanmar Environmental Sustainability Support



Association, and the river fund of the river foundation, Japan. Weitian Ding would
like to take this opportunity to thank the "Nagoya University Interdisciplinary Frontier
Fellowship" supported by Nagoya University and JST, the establishment of university
fellowships towards the creation of science technology innovation, Grant Number
JPMJFS2120.

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





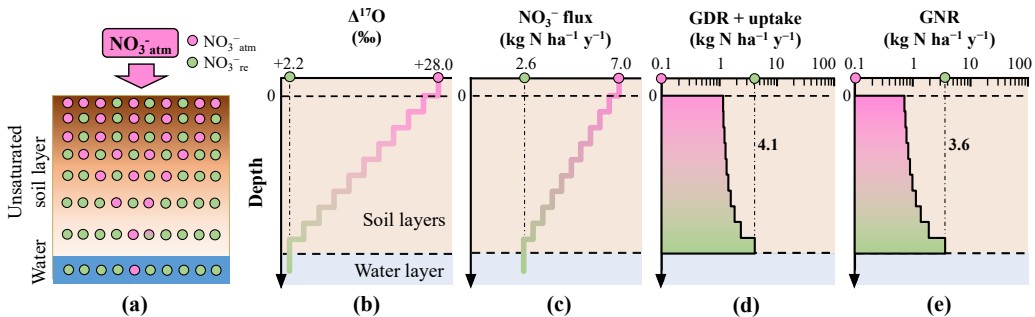

**Figure. 1**. Distribution of $NO_3^-{}_{atm}$ in the simulated forested soil where the distribution
of the $\Delta^{17}O$ values of $NO_3^-$ is heterogeneous (a). Vertical distribution of the following
parameters in the forested soil: the simulated $\Delta^{17}O$ values of $NO_3^-$ (b), simulated
leaching flux of $NO_3^-$ (c), estimated $NO_3^-$ consumption rate (GDR + uptake) (d), and
estimated GNR (e).

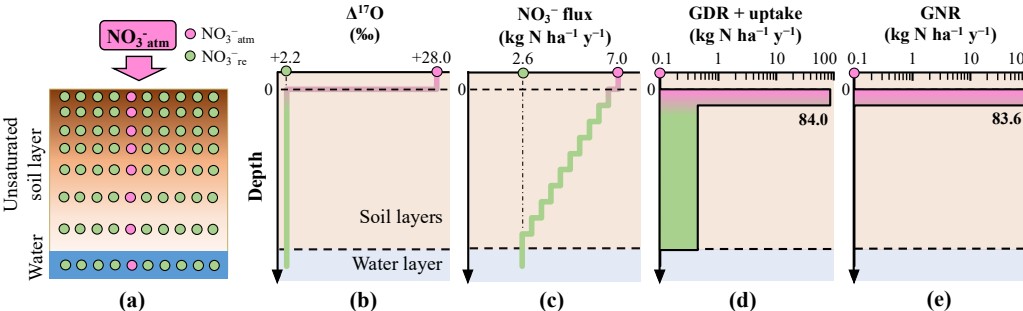

**Figure. 2**. Distribution of $NO_3^-{}_{atm}$ in the simulated forested soil where the distribution
of the $\Delta^{17}O$ values of $NO_3^-$ is homogeneous (a). Vertical distribution of the following
parameters in the forested soil: the simulated $\Delta^{17}O$ values of $NO_3^-$ (b), simulated
leaching flux of $NO_3^-$ (c), estimated $NO_3^-$ consumption rate (GDR + uptake) (d), and
estimated GNR (e).