# Peer review of "Ideas and perspectives: Errors associated with the gross nitrification rates in"

_Biogeosciences, 2022_

## Author Comment (AC1)

**Dear Referee #1**

Thank you very much for your valuable comments on our manuscript. We would like to respond to each of your comments one by one.

First, it is not necessary to assume that the distribution of 17O of nitrate along the soil profile is homogeneous or heterogeneous when apply the nitrate oxygen isotope method to forest soils. In fact, the assumption of the method is that the plants or microbes access the same nitrate source in forest soil with denitrifiers (Fang et al., 2015, PNAS). This assumption is not identical to the assumption that the spatial distribution of 17O of nitrate along the soil profile is homogeneous, as demonstrated by the authors (Fig. 2).

Your understanding of the nitrate isotope method is wrong. The study by Fang et al. (2015) (and other subsequent studies listed in our submitted manuscript) estimated the GNR of forested catchments using Eq. 6:

GNR = NO3-deposition × ( $\Delta^{17}O(NO_3^{-})_{atm} - \Delta^{17}O(NO_3^{-})_{stream}$ )/ $\Delta^{17}O(NO_3^{-})_{stream}$  (6) where  $\Delta^{17}O(NO_3^{-})_{atm}$ ,  $\Delta^{17}O(NO_3^{-})_{stream}$ , and NO3-deposition denote the  $\Delta^{17}O$  value of NO3-atm deposited onto each catchment, the  $\Delta^{17}O$  value of NO3- eluted from each catchment (stream NO3-), and the deposition flux of NO3- into each catchment, respectively. To estimate GNR for each forested catchment using Eq. 6,  $\Delta^{17}O(NO_3^{-})_{stream}$  should be equal to  $\Delta^{17}O(NO_3^{-})_{uptake}$  (the  $\Delta^{17}O$  value of NO3- assimilated by plants) and  $\Delta^{17}O(NO_3^{-})_{denitrification}$  (the  $\Delta^{17}O$  value of NO3- decomposed through denitrification), as explained in our submitted manuscript. That is, the studies that used Eq. 6 to estimate the GNR assumed Eq. 4 in the investigated forested catchments.

 $\Delta^{17}O(NO_3^{-})_{stream} = \Delta^{17}O(NO_3^{-})_{uptake} = \Delta^{17}O(NO_3^{-})_{denitrification}$ (4)

Because most metabolic reactions (GDR + uptake) of NO3- occurred in soil layers within forested catchments, the studies that estimated GNR using Eq. 6 assumed that soil NO3- was homogeneous at the  $\Delta^{17}$ O values and equal to  $\Delta^{17}O(NO_3^{-})_{stream}$ . None of them disclosed this assumption in their papers. Therefore, to clarify this assumption together with its significant impact on the final GNR estimated, we presented GNRs estimated for a forested catchment in which the  $\Delta^{17}O$  values of NO3- in soil layers had been clarified.

**Second, it is not correct to assume that the distribution of 17O of nitrate along the soil profile is homogeneous (Fig. 2). 17O has been rarely measured along the soil profile.**

We agree that the  $\Delta^{17}$ O values of soil NO3- are not always homogeneous in forested catchments. Nevertheless, the study by Fang et al. (2015) and the subsequent studies used Eq. 6, in which the  $\Delta^{17}$ O values of soil NO3- were assumed to be homogeneous

and always equal to  $\Delta^{17}O(NO_3^{-})_{stream}$ , to estimate GNR in each forested catchment, as explained above. As a result, we submitted this manuscript to disclose this critical issue in their estimated GNRs.

**The one and only study shows a sharp decrease in 170 of nitrate in the top soil and remains relatively constant in the soil from 25 to 95 cm (Hattori et al., 2019, Sci. Total Environment). Thus, the assumption made by the author was not supported the field observation.**

First, the accurate vertical distribution of downward NO3- flux had not been clarified in the forested catchment because Hattori et al. (2019) did not monitor the downward water flux in the forested catchment. Thus, in the submitted manuscript, we did not propose any model that represents an accurate vertical distribution of both downward NO3- flux and  $\Delta^{17}$ O values of soil NO3- in the forested catchment studied by Hattori et al. (2019). The linear variation model for both downward flux and the  $\Delta^{17}$ O values of soil NO3- adopted in the simulated calculation is one of the possible variations in soil NO3- in the forested catchment studied by Hattori et al. (2019). The homogenous model could be the case as well. Still, the  $\Delta^{17}$ O data of soil NO3- reported by Hattori et al. (2019), in which >80% of soil nitrate showed  $\Delta^{17}$ O values higher than those in the stream eluted from the catchment (+2.2‰ on average; Fig. 1), implying that the estimated GNR using Eq. 6, in which the  $\Delta^{17}$ O (NO3-)stream, was most likely inaccurate in the forested catchment.

**Figure 1.** Vertical distribution of  $\Delta^{17}$ O values of NO3- in precipitation, each soil layer (0 cm, 25 cm, 55 cm, and 90 cm), ground water, and stream water. Open symbols denote values for summer (June–September), and solid figures denote values for winter (January–April). Box-plot black lines indicate the mean values. Box-plot Box-plot lower and upper boundaries indicate the lower (25%) and upper (75%) quartiles of data in each component, respectively. Whiskers denote the minimum and maximum values reported in each component. The white arrow represents the flux of NO3- inputs and outputs in the ecosystem (Cited from Hattori et al., 2019).

**Third, I agree that it is the distribution of 17O of nitrate along the soil profile is highly hetrogeneous, as nitrification is dominant in surface soils, and deposited nitrate may enter soil from the forest floor. However, it is not correct to assume that 17O of nitrate decreased linearly with soil depth (Fig. 1).**

The linear variation in the  $\Delta^{17}$ O values and the downward flux of soil NO3- in the simulated calculation are possible variations in soil NO3- in the forested catchment. It is impossible to decide whether the linear variation model was correct until the downward water flux, together with the concentration and  $\Delta^{17}$ O values of soil NO3-, is determined for each soil layer, as previously explained.

**The field observation by Hattori et al did not support this assumption. This may be main reason for unrealstically low gross nitrification rate (13 kg N/ha.yr) as calculated by the authors, in the study forest with modate to high N deposition (16 kg N.ha.yr). Nitrification must be strongly active in this forest, which was supported by high soil nitrate concentrations and a large seasonal variation in 15N and 17O of nitrate (Fig. 3 of Hattori et al.).**

Again, the linear variation in the  $\Delta^{17}$ O values and the downward flux of soil NO3- in the simulated calculation are one of the possible variations in soil NO3- in forested catchments. The estimated GNR (13.0 kg of N ha-1 y-1) was also one of the possible values. It is impossible to decide whether the linear variation model was realistic until the downward water flux, together with the concentration and  $\Delta^{17}$ O values of soil NO3-, is determined for each soil layer, as previously explained.

Concerning the concentrations of soil nitrate in the forested catchment reported by Hattori et al. (2019), note that the concentration was not so high, at least in Japanese forested catchments. While the mean concentrations of soil nitrate ranged from 0.8 to 2.3 mg/L (from 13 to 37 uM; Fig. 2) in the study by Hattori et al. (2019), they were 398 uM in the KJ catchment (Nakagawa et al., 2018) and 51 uM in the Matsuzawa catchment (Osaka et al., 2010). Values of >100 uM have also been reported in OYS-O, OYS-M, and TM catchments in Japan (Fang et al., 2015).

**Figure 2.** Vertical distribution of  $NO_3^-$  concentration in precipitation, each soil layer (0 cm, 25 cm, 55 cm, and 90 cm), ground water, and stream water. Box-plot black lines indicate the mean values (Cited from Hattori et al., 2019).

In fact, the nitrate oxgen isotope method admit high heterogeneity of soil nitrfication in both spatially and seasonally. And it is difficult and almost impossible to capture these heterogeneities. However, these heterogeneities can be integrated to streamwater. The nitrate oxygen isotope method takes this advantage of it.

Your understanding of the nitrate isotope method is wrong. As presented in the submitted manuscript, the deviations in the  $\Delta^{17}$ O values of NO3- consumed actually through metabolic reactions (especially uptake reactions) from the mean  $\Delta^{17}$ O values of stream NO3- significantly impacted the estimated GNR. Considering that most of the root biomass is concentrated in the top 10 cm of soils in forested catchments (Jackson et al., 1996), most uptake reactions should occur at the top 10 cm of soil layers and not in the stream. Therefore, possible heterogeneity in the  $\Delta^{17}$ O values of soil NO3- in forested catchments should be considered in estimating GNR. Furthermore, although oxygen isotopes of soil NO3- in forested catchments have rarely been measured along the soil profile, all studies (in which oxygen isotopes were determined for both soil NO3- and stream NO3- simultaneously) showed that the oxygen isotopes of soil NO3- (Hattori et al., 2019; Osaka et al., 2010; Nakagawa et al., 2018), implying that the GNRs of forested catchments estimated using Eq. 6 were inaccurate.

Using Eq. 6 to estimate GNR in forested catchments, it is necessary to verify that the  $\Delta^{17}$ O values of soil NO3- were always equal to those of stream NO3- or to estimate the possible range of errors.

We would like to thank you for the helpful comments. We hope that our responses to your comments are satisfactory.

Sincerely, Weitian Ding PhD student Graduate School of Environmental Studies, Nagoya University Furo-cho, Chikusa-ku, Nagoya, 464-8601, JAPAN Phone: +81-70-4436-3157 E-mail: ding.weitian.v2@s.mail.nagoya-u.ac.jp Cc: Drs. Urumu Tsunogai and Fumiko Nakagawa Reference

Fang, Y., Koba, K., Makabe, A., Takahashi, C., Zhu, W., Hayashi, T., Hokari, A. A., Urakawa, R., Bai, E., Houlton, B. Z., Xi, D., Zhang, S., Matsushita, K., Tu, Y., Liu, D., Zhu, F., Wang, Z., Zhou, G., Chen, D., Makita, T., Toda, H., Liu, X., Chen, Q., Zhang, D., Li, Y. and Yoh, M.: Microbial denitrification dominates nitrate losses from forest ecosystems, Proc. Natl. Acad. Sci. U. S. A., 112(5), 1470–1474, doi:10.1073/pnas.1416776112, 2015.

Hattori, S., Nuñez Palma, Y., Itoh, Y., Kawasaki, M., Fujihara, Y., Takase, K. and Yoshida, N.: Isotopic evidence for seasonality of microbial internal nitrogen cycles in a temperate forested catchment with heavy snowfall, Sci. Total Environ., 690, 290– 299, doi:10.1016/j.scitotenv.2019.06.507, 2019.

Jackson, R. B., Canadell, J., Ehleringer, J. R., Mooney, H. A., Sala, O. E. and Schulze, E. D.: A global analysis of root distributions for terrestrial biomes, Oecologia, 108(3), 389–411, doi:10.1007/BF00333714, 1996.

Nakagawa, F., Tsunogai, U., Obata, Y., Ando, K., Yamashita, N., Saito, T., Uchiyama, S., Morohashi, M. and Sase, H.: Export flux of unprocessed atmospheric nitrate from temperate forested catchments: A possible new index for nitrogen saturation, Biogeosciences, 15(22), 7025–7042, doi:10.5194/bg-15-7025-2018, 2018.

Osaka, K., Ohte, N., Koba, K., Yoshimizu, C., Katsuyama, M., Tani, M., Tayasu, I. and Nagata, T.: Hydrological influences on spatiotemporal variations of  $\delta^{15}$ N and  $\delta^{18}$ O of nitrate in a forested headwater catchment in central Japan: Denitrification plays a critical role in groundwater , J. Geophys. Res. Biogeosciences, 115(G2), n/a-n/a, doi:10.1029/2009jg000977, 2010.

---

## Author Comment (AC2)

Dear Referee #2

Thank you very much for your valuable comments and questions on our manuscript. We would like to respond to each of your comments one by one.

**I suggest that Ding et al. elaborate on the assumptions of their simulation. For example, is it realistic to assume that D17O decreases linearly with depth? In this scenario, the nitrate (NO3) flux down the soil profile would consist of mostly unprocessed atmospheric NO3 (at least until halfway down the depth profile). This does not appear to align with the empirical data used in the simulation from Hattori et al.**

Thank you for your question. First, note that the linear variation model used in the simulated calculation is one of the possible variations in the $\Delta^{17}O$ values of soil $NO_3^-$ in forested catchments. It is impossible to decide whether the linear variation model was realistic until the downward water flux, together with the concentration and $\Delta^{17}O$ value of $NO_3^-$, is determined for each soil layer.

However, the simultaneous observations of the oxygen isotopes of soil $NO_3^-$ and stream $NO_3^-$ (Hattori et al., 2019; Osaka et al., 2010; Figs. 1 and 2) implied that the homogeneous model assumed in past studies, in which the $\Delta^{17}O$ values of soil $NO_3^-$ were homogeneous and always equal to the mean $\Delta^{17}O$ value of stream $NO_3^-$ as we explained in the submitted manuscript, was unrealistic. This is why we proposed the linear variation model as a possible alternative model for the simulated calculation.

[Figure]

**Figure 1.** Vertical distribution of $\Delta^{17}O$ values of $NO_3^-$ in precipitation, each soil layer (0 cm, 25 cm, 55 cm, and 90 cm), ground water, and stream water (Cited from Hattori et al., 2019).

[Figure]

**Figure 2.** Vertical distribution of $\delta^{18}O$ values of $NO_3^-$ in the rain, each soil layer (0 cm, 10 cm, 20 cm, 30 cm, and 50 cm), ground water, and stream water (Cited from Osaka et al., 2010).

**How would the simulation results change if D17O dramatically decreased from deposition to the first soil depth used in the simulation? Existing data show that D17O of NO3 can be low in the uppermost soil horizons (<5 per mille in the upper 7 cm, Yu and Elliott 2018; 2 per mille +/- 1.1 per mille from 0-30 cm, Costa et al., 2011), which suggests that the assumption of a linear decrease of D17O from deposition to "water" (i.e., the lowest soil depth in the simulation; Figure 1) is not necessarily valid.**

Thank you for your comment. In response to your request, we made a new simulated calculation in which the $\Delta^{17}O$ values of $NO_3^-$ were decreased from +28.0 ‰ to +5.0 ‰ at the first layer and then gradually decreased with the subsequent nine layers, from +5.0 ‰ to +2.2 ‰ with a decrease rate of +0.31 ‰ for each step (Table 1). As a result, the simulated GNR (36.1 kg of N ha$^{-1}$ y$^{-1}$) was significantly smaller than the GNR (83.6 kg of N ha$^{-1}$ y$^{-1}$) calculated using Eq. 6, in which the $\Delta^{17}O$ values of $NO_3^-$ in the forested soils were homogeneous.

**Table 1.** $\Delta^{17}O$ values of $NO_3^-$, leaching flux of $NO_3^-$, total metabolic rate of $NO_3^-$ (GDR + uptake), and GNR in the simulated forested soil where the distribution of $\Delta^{17}O$ values of $NO_3^-$ is heterogeneous with the values decreased from +28.0 to +5.0 ‰ at the first layer, and gradually decreased with the later 9 layers, from +5.0 ‰ to +2.2 ‰.

| Depth layer | $\Delta^{17}O$ ‰ | $NO_3^-$ flux | GDR +uptake | GNR |
|---|---|---|---|---|
| | | kg of N ha$^{-1}$ y$^{-1}$ | | |
| 0 | 28.0 | 7.0 | 0.0 | 0.0 |
| 1 | 5.0 | 6.6 | 32.6 | 32.2 |
| 2 | 4.7 | 6.1 | 0.9 | 0.4 |
| 3 | 4.4 | 5.7 | 0.9 | 0.4 |
| 4 | 4.1 | 5.2 | 0.9 | 0.4 |
| 5 | 3.8 | 4.8 | 0.9 | 0.4 |
| 6 | 3.4 | 4.4 | 0.9 | 0.4 |
| 7 | 3.1 | 3.9 | 0.9 | 0.4 |
| 8 | 2.8 | 3.5 | 0.9 | 0.4 |
| 9 | 2.5 | 3.0 | 0.9 | 0.4 |
| 10 | 2.2 | 2.6 | 0.9 | 0.4 |
| 11 | 2.2 | 2.6 | 0.0 | 0.0 |
| Total | | | | 36.1 |

**I also request the authors elaborate on the assumption that there is no biological processing in the water layer. Is this water layer the stream? Or is it intended to be the saturated zone of the soil? If it is the latter, is it reasonable to assume there is no processing along flowpaths between the saturated zone and the stream? How would the simulation results change if this assumption was violated?**

The name of the final layer does not influence the estimated GNR. What we called the "water layer" in the submitted manuscript was the layer where the biological processing of $NO_3^-$ was negligible. If the biological processing was significant in the water layer, it can be integrated into the other layers where biological processing was active.

**While Ding et al. present a simulation that suggests inaccurate GNR estimates produced by the triple oxygen isotope approach at the watershed scale, I suggest the authors expand the scope of their premise to make it more broadly relevant. GNR, as calculated by soil scientists in soil cores (which has been done with D17O, Yu and Elliott 2018, along with many others using d15N), are not equal to gross nitrification rates as calculated at the watershed scale using streamwater NO3. I think this difference needs to be made clear and this manuscript represents a potential outlet to do so.**

Thank you for your comments. Yu and Elliott (2018) sampled various soil cores for incubating with fertilizer $NO_3^-$ enriched in $^{17}O$ to estimate GNR using the isotopic mass balance. Because the $\Delta^{17}O$ values of soil $NO_3^-$ in the incubation bottles were homogeneous, the estimated GNR should be relevant. However, $\Delta^{17}O$ values of soil $NO_3^-$ on the catchment scale were heterogeneous and different from the mean $\Delta^{17}O$ value of stream $NO_3^-$, as explained in our manuscript. Thus, the GNRs estimated on the catchment scale by the past studies were most likely inaccurate; hence, we should consider that significant errors were included in the estimated GNRs. We would like to add this to the revised manuscript. Thank you.

**For example, if D17O is a conservative tracer, a decrease in D17O of NO3 between deposition and streamwater requires addition of or dilution by terrestrial NO3, which has D17O = 0 (or approximately equal to 0). By this logic, Hattori's D17O data indicate that the streamwater NO3 measured must have had terrestrial NO3 added (i.e., nitrification) or have been diluted by terrestrial NO3 along its flowpath from deposition to stream. I think this example illustrates the difference between GNR as measured in a soil core and GNR as measured at the watershed scale. Perhaps it would be more appropriate to give a different name to the watershed scale metric that can be calculated using eq. 6.**

We could obtain an accurate estimated GNR of the forested catchment scale if we estimated the downward water flux and the concentration and $\Delta^{17}O$ value of $NO_3^-$ in each soil layer, as presented in the submitted manuscript. If we can obtain such an accurate catchment scale GNR, the GNR will be comparable to that estimated from the core scale and then integrated into the catchment scale.

**Specific comments:**

**-Lines 16-17: needs reference. Is NO3 often limiting? Or NH4? Or just N more generally?**

Thank you for your comment. We would like to revise the sentence here as follows.
Nitrate is one of the important nitrogen nutrients for primary production in forested ecosystems.

**-Line 24: I suggest a different word than "determined". Perhaps "quantified" or "estimated"**

Thank you for your comment. We would like to use "estimated" here.

**-Line 24: What is a "water environment"? A stream? A pond? A lake? Soil water?**

Thank you for your comment. We would like to use "lake" here.

**-Line 30-31: I suggest providing a range of D17O values of atmospheric NO3. 26 per mille is a good average, but it can be much lower.**

Thank you for your comment. We would like to add the range of $\Delta^{17}O$ values of atmospheric $NO_3^-$ here.

**-Line 32: What does "almost stable" mean?**

Thank you for your question. It is not stable in the strict sense; however, it is stable on the per mil scale.

**-Eq. 1: I suggest a different subscript for "water". Readers might see "water" and think you are measuring the triple oxygen isotopes of O in H2O.**

Thank you for your question. We would like to use "$\Delta^{17}O$" here.

**-Line 46: Riha et al. 2014 was not a forested catchment. It was an urban watershed study.**

Thank you for your comment. We would like to revise the reference.

**-Line 54: reference for NO3 being homogenous?**

Thank you for your question. We would like to add the references (Tsunogai et al., 2011, 2018).

**-Lines 95-99: Where were the soil data collected in relation to the stream for Hattori et al.?**

Sampling points of soil water (shaded rectangle) and stream water (white inverted triangle) were shown in Fig. 3.

[Figure]

**Figure 3.** Map and sampling points (Cited from Hattori et al., 2019).

**Is it reasonable to assume there was no processing along the flow paths from soil to stream?**

Because the downward water flux has not been determined for each soil layer in the forested catchment, the accurate $NO_3^-$ processing along the flow path from soil to stream is unknown. Again, the linear variation model is one of the possible variation models. We would like to emphasize this in the revised manuscript.

**-Line 136: Are you missing a word at the end of this sentence? Perhaps add "sources" to the end.**

Thank you for your question. We would like to revise the sentence as follows.

If we estimated the downward water flux at each soil layer, together with the concentration and $\Delta^{17}O$ value of $NO_3^-$ in each soil layer using a tension-free lysimeter (Inoue et al., 2021),we could estimate the vertical changes in the leaching flux of $NO_3^-$ for each soil layer together with the $\Delta^{17}O$ value of $NO_3^-$ in each soil layer.

We would like to thank you for the helpful comments and questions. We hope that our responses to your comments are satisfactory.

Sincerely,
Weitian Ding
PhD student
Graduate School of Environmental Studies,
Nagoya University
Furo-cho, Chikusa-ku, Nagoya,
464-8601, JAPAN
Phone: +81-70-4436-3157

E-mail: ding.weitian.v2@s.mail.nagoya-u.ac.jp
Cc: Drs. Urumu Tsunogai and Fumiko Nakagawa

**Reference**

Hattori, S., Nuñez Palma, Y., Itoh, Y., Kawasaki, M., Fujihara, Y., Takase, K. and Yoshida, N.: Isotopic evidence for seasonality of microbial internal nitrogen cycles in a temperate forested catchment with heavy snowfall, Sci. Total Environ., 690, 290–299, doi:10.1016/j.scitotenv.2019.06.507, 2019.

Inoue, T., Nakagawa, F., Shibata, H., and Tsunogai, U.: Vertical Changes in the Flux of Atmospheric Nitrate From a Forest Canopy to the Surface Soil Based on $\Delta^{17}O$ Values, J. Geophys. Res.-Biogeo., 126, e2020JG005876, https://doi.org/10.1029/2020JG005876, 2021.

Osaka, K., Ohte, N., Koba, K., Yoshimizu, C., Katsuyama, M., Tani, M., Tayasu, I. and Nagata, T.:    Hydrological influences on spatiotemporal variations of $\delta^{15}N$ and $\delta^{18}O$ of nitrate in a forested headwater catchment in central Japan: Denitrification plays a critical role in groundwater , J. Geophys. Res. Biogeosciences, 115(G2), n/a-n/a, doi:10.1029/2009jg000977, 2010.

Tsunogai, U., Daita, S., Komatsu, D. D., Nakagawa, F. and Tanaka, A.: Quantifying nitrate dynamics in an oligotrophic lake using $\Delta^{17}O$, Biogeosciences, 8(3), 687–702, doi:10.5194/bg-8-687-2011, 2011.

Tsunogai, U., Miyauchi, T., Ohyama, T., Komatsu, D. D., Ito, M. and Nakagawa, F.: Quantifying nitrate dynamics in a mesotrophic lake using triple oxygen isotopes as tracers, Limnol. Oceanogr., 63, S458–S476, doi:10.1002/lno.10775, 2018.

Yu, Z. and Elliott, E. M.: Probing soil nitrification and nitrate consumption using $\Delta^{17}O$ of soil nitrate, Soil Biol. Biochem., 127(June), 187–199, doi:10.1016/j.soilbio.2018.09.029, 2018.